# Comparative Experimental Effects of Intercropping and Cypermethrin on Insect Pest Infestation and Yield of Maize, Cowpea and Okra in Two Cameroonian Agro-Ecological Zones

**Patient Farsia Djidjonri [1],\*, Nukenine Elias Nchiwan [1] and Hartmut Koehler [2]**

[1] Department of Biological Sciences, University of Ngaoundéré, 455 Ngaoundere, Cameroon; elinchiwan@yahoo.fr

[2] Center for Environmental Research and Sustainable Technology, University of Bremen, D 28359 Bremen, Germany; hartmut.koehler@uni-bremen.de

\* Correspondence: djidjonri@yahoo.fr

**Abstract:** The present study investigates the effect of intercropping (maize-cowpea, maize-okra, maize-okra-cowpea, okra-cowpea) compared to insecticide application on the level of infestation of insect pests and the final yield of maize, cowpea and okra. Field experiments were conducted during the 2016 and 2017 cropping seasons in the Guinean Savannah (Dang-Ngaoundere) and Sudano Sahelian (Gouna-Garoua) agro-ecological zones in Cameroon. Our experimental design was a split plot arrangement in a randomized complete block with four replications. The main factor was assigned to the use of insecticide (Cypermethrin) and sub plots were devoted for cropping systems. We compared the efficiency of intercropping to that of Cypermethrin application on the Yield of maize, cowpea and okra as influenced by insect pest damages. The comparison of monocropped sprayed by Cypermethrin to unsprayed showed that, in Dang, insect pests reduced maize yield by 37% and 24% in 2016 and 2017, respectively, whereas in Gouna, it was lower than 8% during the both years. Reduction in seed yield by insect pests on cowpea in Dang represented 47% and 50% in 2016 and 2017, respectively, whereas in Gouna, it was 55% and 63% in 2016 and 2017, respectively. For okra, insect pests reduced okra fruit yield by 25% and 44% in Dang and 23% and 28% in Gouna, respectively, in 2016 and 2017. Crop yield was lower in intercropping compared to monoculture due to competition of plants in association on different resources. Considering the total yields obtained from each intercropping, intercropping trials resulted generally in higher yields compared to monoculture (LER > 1) in both sites and years but the respective yields were quite different. On the basis of the results obtained, we recommend maize-cowpea intercropping as a sustainable solution to reduce the infestation level of their pest insects.

**Keywords:** intercropping; cypermethrin; insect; pest; productivity; Cameroon

## 1. Introduction

The loss of crop due to insect pests remains quite high worldwide. Insect pests are reported to be responsible for destroying 20% of the world's total crop production annually [1]. The fight against insect pests remains a matter of concern for farmers. About 3.5 billion kg of chemical pesticides are used annually in the world [2]. This represents an annual investment of 40 billion US dollars that cannot be used by farmers for their health and others. Moreover, chemical insecticides are reported to be unfriendly to the environment and pose health hazards to humans which result from, among others things, their neurotoxicity, reproductive toxicity, and cancerogeneity [3–6]. Intercropping has been proposed as a solution to prevent the environment and farmers from the effects of chemical pesticides [7–9].

The practicability and efficiency of intercropping has been shown in many studies [10–13]. The efficiency of intercropping results from an increase of habitat and diversity

of organisms. This again leads to complex interactions, including the control of insect gradations leading to unwanted crop losses. Study in Kenya showed that when maize, nappier grass and Desmodium are intercropped, Desmodium produce aromas that repel stem borer, while nappier grass emits aromas that encourage moths to lays their eggs on them and produce gummy substance that traps the larvae after their hatching, reducing their number [14]. Moreover, insect pests were reported to settle on crops only when host factors such as visual stimulus, taste and smell are satisfied, and this is more likely in monocultures where the chances of meeting a wrong stimulus is lower [15].

Our study focuses on two agro-ecological zones in Cameroon with unimodal rainfall, the Guinean Savannah and Sudano Sahelian zone, in Adamawa and the North, respectively. Here, farmers cultivate maize in association with cowpea and okra, but according to our knowledge, there have been no scientific studies to test the efficiency of these intercropping systems against the level of insect pest infestation and crop productivity. The present study aims to investigate the effect of intercropping (maize-cowpea, maize-okra, maize-okra-cowpea and okra-cowpea) on the level of infestation of insect pests and compare the efficiency of intercropping to that of insecticide application on the yield of maize, cowpea, and okra for the respective trials.

## 2. Materials and Methods

### 2.1. Study Sites

Studies were conducted in 2016 and 2017 in the mid and late rainy season and the beginning of dry season (harvest; June to November) in Dang-Ngaoundéré (07°24′08.2″ N; 13°33′01.6 E; 1094 ASL; Guinean Savannah agro-ecological zone) and Gouna-Garoua (08°29′53.4″ N; 13°31′00.3 E; 402 ASL, Sudano Sahelian agro-ecological zone) of Adamawa and North, respectively, Cameroon. The sites selected for our field experiments differ in many factors, such as climate, soil, and vegetation.

### 2.2. Precipitation, Temperature, Soil, and Vegetation

Weather conditions differed greatly among the both localities [16]. The annual precipitation was 1352 mm (2016) and 1400 mm (2017) in Ngaoundéré, 999 mm (2016) and 789 mm (2017) in Garoua. The rainy season begins in Ngaoundéré and Garoua in March and April, respectively, then ends in October in both localities. Average annual temperatures were 23.13 °C (2016) and 23.09 °C (2017) in Ngaoundéré, 30.74 °C (2016) and 30.02 °C (2017) in Garoua. The soil of Dang is compact brown-reddish, developed on basaltic rock, with pH 4.38 (Table 1). The surroundings of the study site are agricultural or fallow, with wild and cultivated species like maize, cowpea, groundnut, millet, bean, potatoes, tomatoes, pepper, and ginger. Gouna soil is ferruginous, well drained with pH 5.03 (Table 1). The vegetation near the study site was represented by wild and cultivated species like maize, cowpea, cotton, groundnut, millet, and bean and wild.

**Table 1.** Soil chemical composition.

| Locality | $PO_4^{3-}$ mg/ 100 g Soil | Mg mg/ 100 g Soil | Ca mg/ 100 g Soil | K mg/ 100 g Soil | pH | Total C/N | Total N% | Total C% |
|---|---|---|---|---|---|---|---|---|
| Gouna | 0.36 | 6.88 | 46.47 | 10.87 | 5.03 | 20.30 | 0.03 | 0.67 |
| Dang | 0.55 | 15.80 | 78.55 | 16.92 | 4.38 | 15.73 | 0.16 | 2.47 |

### 2.3. Experimental Set-Up, Sowing and Weeding

The experimental design was a split plot with two factors. The main factor was assigned for the application of Cypermethrin and the sub plots were devoted for cropping systems. Size of each sub plot was 4.5 m × 3.2 m, with 1 m spacing. We implanted the seven cropping systems (Table 2) in four replications without and with Cypermethrin application, resulting in a total of fifty-six plots per site. The spacing in planting was according to traditional farming: maize in monoculture row spacing with 80 cm and 50 cm between

plants in the row; cowpea in monoculture row spacing with 80 cm and 30 cm between cowpea plants; okra in monoculture row spacing with 80 cm and 40 cm between plants in the row. For maize/cowpea, maize/okra, and okra/cowpea, the first plant was sown as in monoculture and the seeds of the second intercropped plant were seeded in the middle of the lines of the first plant. For maize-cowpea-okra intercropping, maize was sown as in monoculture; the okra seeds were planted at 30 cm and the cowpea seeds at 55 cm from the first line of maize. For all treatments, 3–4 seeds of maize, 3–4 seeds of cowpea, and 4–6 seeds of okra were sown per hole. After germination (two weeks), the plots were thinned to one plant per stand for maize and okra plants and to two plants per stand for cowpea. Weed control was done 3, 7, and 10 weeks after planting by hand using a hoe.

**Table 2.** Cropping systems implanted in the experimental fields.

| Monoculture | Intercropping |
| :---: | :---: |
| **Maize** | Maize + Cowpea |
| **Cowpea** | Maize + Okra |
| **Okra** | Okra + Cowpea |
| | Maize + Okra + Cowpea |

All without and with Cypermethrin.

### 2.4. Fertilizer Amendment and Cypermethrin Application

NPK 15:15:15 at the rate of 150 kg ha$^{-1}$ was applied to all plots except cowpea monocropped plots two weeks after sowing [17]. Seventy days after planting, urea fertilizer was applied to all plots except cowpea monocropped plots, each at the rate of 50 kg ha$^{-1}$.

Spraying of the positive control plot with Cypermethrin 100 EC started two weeks after sowing until the crops matured with intervals of one week. As recommended by manufacturer, a dose of 500 mL ha$^{-1}$ was used.

### 2.5. Yield Parameters

2.5.1. Maize

The method of Ijoyah & Dzer [18] was used to assess maize yield. Maize was harvested when the leaves turned yellowish and had fallen off, which were signs of senescence and cob maturity. Forty plants were randomly selected within each treatment, cobs were counted, and the average was taken as the number of cobs per plant. Ten of the cobs were later shelled manually, the seeds removed and counted, and the average was taken as the number of seeds per cob. Afterwards, 1000 seeds from each treatment were weighed using an electronic balance. The result was used to obtain the yield in t ha$^{-1}$.

2.5.2. Cowpea

The yield components were determined at harvest when the crops on the field were sufficiently dry. The determination of Cowpea seeds yield was done by the method of Gandébé et al. [19]. These yield parameters were the number of pods per plant (NPPt), which was quantified by counting and the average determined. The number of seeds per pod (NSPd) was determined by threshing 40 randomly selected pods from the sampled plants. Then, the number of seeds per plant (NSPt) was obtained as the product, NPPt × NSPd. The 100-seed weight (100-SW) was evaluated by weighing a random sample seeds from the experimental unit using an electronic balance. Seed yield was then quantified as the product (NSPt × 100-SW). Grain yield was converted to hectares (kg ha$^{-1}$).

2.5.3. Okra

Harvest of the fresh fruits started when fruits were easy to break with a finger and it was done twice a week. Forty plants were randomly selected within each treatment and the number of fruits were determined. These same fruits were weighed using an electronic balance. The result was used to obtain the yield in t ha$^{-1}$.

### 2.6. Land Equivalent Ratio

Maize, Cowpea, Okra

The productivity of monocropped and intercropped crops was compared by calculating the land equivalent ratio (*LER*) as described by Mead and Willey [16].

$$LER = \frac{\text{intercrop yield of plant a}}{\text{monocrop yield of plant a}} + \frac{\text{intercrop yield of plant b}}{\text{monocrop yield of plant b}}$$

When the value of *LER* is greater than 1, the intercropping favors the growth and yield of the species, whereas when *LER* is lower than 1, the intercropping negatively affects the growth and yield of crops grown in mixtures [20]. Thus, *LER* gives a good assessment of the competitive abilities of the respective crops. It also indicates the yield advantage of intercropping.

### 2.7. Data Analysis

Data obtained were tested for normality. The Tukey test were used for the separation of more than two means, using the SPSS software 16.0. The Student *t*-test was used for comparison of the means of two samples.

## 3. Results and Discussion

### 3.1. Yield

3.1.1. Maize

Table 3 presents the maize seed yield as influenced by Cypermethrin treatments and the intercropping system in 2016 and 2017 at Dang and Gouna. Overall, maize seed yield was higher at Gouna than at Dang during the both cropping seasons. This difference might be the result of lower soil pH and higher insect pest infestation observed at Dang. This result confirms the fact that the North region ranks above the Adamawa region in the production of maize in Cameroon [21]. Maize seed yield was affected by cropping system and Cypermethrin in both sites.

**Table 3.** Maize yield (t ha$^{-1}$) as influenced by Cypermethrin treatments and intercropping system in 2016 and 2017 at Dang and Gouna.

| Treatments | Dang 2016 | | | Dang 2017 | | |
|---|---|---|---|---|---|---|
| | **Unsprayed** | **Sprayed** | *t* | **Unsprayed** | **Sprayed** | *t* |
| M | 2.31 ± 0.12 [ab] | 3.67 ± 0.17 [a] | 6.60 *** | 2.27 ± 0.26[a] | 2.99 ± 0.33 [a] | 7.31 *** |
| MC | 2.60 ± 0.13 [a] | 3.03 ± 0.16 [b] | 2.17 * | 2.27 ± 0.04 [a] | 2.64 ± 0.03 [b] | 7.34 *** |
| MO | 2.23 ± 0.09 [ab] | 2.58 ± 0.14 [bc] | 2.07 * | 2.23 ± 0.08 [a] | 2.62 ± 0.05 [b] | 4.06 *** |
| MCO | 1.86 ± 0.14 [b] | 2.06 ± 0.16 [c] | 1.86 | 1.95 ± 0.06 [b] | 2.17 ± 0.04 [c] | 0.57 |
| Mean | 2.25 ± 0.08 | 2.83 ± 0.08 | 5.27 *** | 2.18 ± 0.03 | 2.60 ± 0.03 | 8.86 *** |
| $F_{(3, 156)}$ | 3.93 ** | 30.16 *** | | 9.07 *** | 27.52 *** | |

| Treatments | Gouna 2016 | | | Gouna 2017 | | |
|---|---|---|---|---|---|---|
| | **Unsprayed** | **Sprayed** | *t* | **Unsprayed** | **Sprayed** | *t* |
| M | 4.62 ± 0.29 [a] | 4.98 ± 0.19 [a] | 1.07 | 4.16 ± 0.19 [a] | 4.37 ± 0.14 [a] | 1.56 |
| MC | 4.14 ± 0.10 [ab] | 4.30 ± 0.13 [b] | 1.03 | 4.06 ± 0.10 [a] | 4.08 ± 0.11 [a] | 0.03 |
| MO | 3.83 ± 0.11 [b] | 3.87 ± 0.11 [b] | 0.23 | 3.40 ± 0.05 [b] | 3.87 ± 0.09 [ab] | 2.29 * |
| MCO | 2.97 ± 0.06 [c] | 3.19 ± 0.13 [c] | 1.73 | 2.91 ± 0.14 [c] | 3.35 ± 0.12 [b] | 2.01 |
| Mean | 3.89 ± 0.08 | 4.08 ± 0.10 | 1.54 | 3.76 ± 0.06 | 3.80 ± 0.08 | 0.42 |
| $F_{(3, 156)}$ | 26.27 *** | 19.70 *** | | 32.48 *** | 13.13 *** | |

M: maize; MO: maize + okra; MC: maize + cowpea; MCO: maize + cowpea + okra. Results are expressed as mean (±) standard error. In column, means followed by same letter are not significantly different at 5%. *: $p < 0.05$; **: $p < 0.01$; ***: $p < 0.001$.

The difference between the maize yields recorded in the sprayed plots and in the unsprayed is associated to insect infestations. Unprotected maize plots at Dang suffered more from the activity of insect pests which results in the decreasing seed yield. In mono-

culture at Dang, insect pests reduced maize yield by 37% and 24% in 2016 and 2017, respectively. At Gouna, the reduction yield due to insect pests was lower than 8% during the both years. The results found in Dang are similar to those reported by Ndemah and Schulthess [22] that yield loss due to prominent stem borers, *Busseola fusca* Fuller, 1901 (Lepidoptera: Noctuidae) and *Sesamia calamistis* Walker, 1865 (Lepidoptera: Pyralidae) ranges between 17 to 44% in Cameroon. A yield loss of 22 to 67% on maize due to the infestation of *Spodoptera frugiperda* (Smith & Abbot, 1797) (Lepidoptera: Noctuidae) was reported in Ghana by Day et al. [23]. Similar results were also reported by Kansiime et al. [24] in Zambia.

During the 2016 cropping season at Dang, among the unsprayed, maize associated to cowpea recorded a slightly higher seed yield compared to maize monocropped, but there was not a significant difference. This finding shows the benefit of maize associated to cowpea to control maize insect pests. These results are in line with those of Okigbo et al. [25], who found the incidence of stem borer pests in the humid forest areas of Cameroon to be lower in maize intercropped with cassava, cowpea, and soybean than when monocropped; and those of Maluleke [26], where maize stem borer was found to be more severe in mono- than in intercropping with lablab. Kfir et al. [27] found that maize mono-crops had three to nine times more damage than maize intercropped with non-host crops such as cowpea, cassava, and soybean, in studies carried out in Cameroon. However, between intercropping trials, maize-cowpea-okra reduced the maize seed significantly compared to maize-cowpea. Concerning the sprayed trials in both years at Dang, maize monocropped recorded a significant higher seed yield compared to maize in intercropping during both seasons. In detail, yields of maize-cowpea, maize-okra and maize-cowpea-okra intercropping represent respectively 82.66%, 70.30%, and 56.13% in 2016 and 88.29%, 87.63%, and 72.52% in 2017 of maize yield recorded in monoculture. In intercropping, no significant difference was observed between the yield recorded on maize-cowpea and maize-okra intercropping.

In Gouna, among the unsprayed, the maize seed yield was statistically the same between maize monocropped and maize associated to cowpea during both years. However, maize-cowpea-okra intercropping reduced the maize yield significantly compared to maize in monoculture in 2016 and additionally maize-okra in 2017. Among the sprayed treatments, maize monocropped recorded a higher seed yield compared to maize in all intercropping trials in both years. Yields of maize-cowpea, maize-okra, maize-cowpea-okra intercropping represent, respectively, 86.35%, 77.71%, 64.06% in 2016; 93.36%, 88.56%, and 76.66% in 2017 of maize yield recorded in monoculture.

The higher grain yield found in maize monocropped sprayed compared to intercropping sprayed could be attributed to the lower plant density and lack of competition for resources such as light, nutrients, and water when growing in monoculture. This result is in agreement with Ijoyah & Dzer [18], who found high cereal grain yield on maize in monoculture than in maize intercropped with okra. In contrast, Mpairwe et al. [28] and Dapaah et al. [29] found a higher grain yield on maize intercropped with soybean and cowpea than in maize monocropped. Similarly, Ogah et al. [30] in Nigeria found higher grain yields on maize intercropped with Bambara groundnut compared to maize in monoculture. The higher seed yield recorded on maize associated with cowpea compared to other intercropping shows the importance of legumes when growing in combination with cereals. A complex series of inter and intraspecific interactions guided by modifications and uses of light, water, nutrients, and enzymes was reported on intercropping between cereals and legumes [31]. Many plants have the ability to modify the rhizosphere pH and improve the availability of nutrients such as P, K, Ca, and Mg, which are in unavailable forms [32].

### 3.1.2. Cowpea

Table 4 presents the cowpea seed yield as influenced by Cypermethrin treatments and intercropping systems. Cypermethrin significantly increased the cowpea seed yield

in comparison to the intercropping system. The comparison of monocropped sprayed to monocropped unsprayed showed that insect pests reduced cowpea seed yield by 47% and 50% in 2016 and 2017, respectively, in Dang. In Gouna, the reduction of yield due to insects was 55% and 63% in 2016 and 2017, respectively. In accordance with our finding, cowpea yield reduction due to insect pests was reported to range from 20 to 80% [33]. The high effectiveness of Cypermethrin to reduce the effect of insect pests on cowpea yield could be associated with its standardized active ingredient formulations that have "knockdown" effects on pests immediately on exposure, like all pyrethroids do [34].

**Table 4.** Cowpea seed yield (t ha$^{-1}$) as influenced by Cypermethrin treatments and intercropping system in 2016 and 2017 at Dang and Gouna.

| Treatments | Dang 2016 | | | Dang 2017 | | |
|---|---|---|---|---|---|---|
| | **Unsprayed** | **Sprayed** | *t* | **Unsprayed** | **Sprayed** | *t* |
| C | 0.87 ± 0.06 [a] | 1.65 ± 0.12 [a] | 5.72 *** | 0.85 ± 0.07 [a] | 1.69 ± 0.14 [a] | 6.15 *** |
| MC | 0.75 ± 0.05 [ab] | 0.95 ± 0.05 [b] | 2.74 * | 0.65 ± 0.06 [ab] | 0.89 ± 0.07 [b] | 2.58 * |
| OC | 0.61 ± 0.04 [b] | 0.88 ± 0.06 [b] | 3.89 *** | 0.63 ± 0.09 [b] | 1.01 ± 0.04 [b] | 3.51 ** |
| MCO | 0.42 ± 0.04 [c] | 0.54 ± 0.03 [c] | 2.38 * | 0.56 ± 0.06 [b] | 0.59 ± 0.03 [c] | 0.44 |
| Mean | 0.66 ± 0.03 | 1.01 ± 0.04 | 6.15 *** | 0.67 ± 0.03 | 1.04 ± 0.05 | 6.12 *** |
| $F_{(3, 156)}$ | 14.91 *** | 39.88 *** | | 5.29 ** | 28.13 *** | |
| Treatments | Gouna 2016 | | | Gouna 2017 | | |
| | **Unsprayed** | **Sprayed** | *t* | **Unsprayed** | **Sprayed** | *t* |
| C | 0.65 ± 0.04 [a] | 1.76 ± 0.09 [a] | 11.54 *** | 0.73 ± 0.04 [a] | 1.64 ± 0.07 [a] | 11.36 *** |
| MC | 0.66 ± 0.05 [a] | 0.96 ± 0.06 [b] | 4.04 *** | 0.76 ± 0.06 [a] | 0.92 ± 0.05 [b] | 2.10 * |
| OC | 0.55 ± 0.04 [ab] | 0.92 ± 0.07 [b] | 4.89 *** | 0.57 ± 0.04 [b] | 0.90 ± 0.05 [b] | 5.22 *** |
| MCO | 0.46 ± 0.05 [b] | 0.58 ± 0.04 [c] | 1.90 | 0.42 ± 0.04 [b] | 0.60 ± 0.04 [c] | 3.09 ** |
| Mean | 0.58 ± 0.02 | 1.05 ± 0.05 | 9.10 *** | 0.62 ± 0.03 | 1.01 ± 0.04 | 8.34 *** |
| $F_{(3, 156)}$ | 5.38 ** | 57.37 *** | | 14.65 *** | 60.30 *** | |

C: cowpea; MC: maize + cowpea; OC: okra + cowpea; MCO: maize + cowpea + okra. Results are expressed as mean (±) standard error. In column, means followed by same letter are not significantly different at 5%. *: $p < 0.05$; **: $p < 0.01$; ***: $p < 0.001$.

In 2016 at Dang, among the unsprayed, cowpea monocropped produced a higher seed yield followed by cowpea in association to maize and cowpea-maize-okra intercropping. Among the sprayed trials, cowpea monocropped produced a significant higher seed yield compared to cowpea in all intercropping trials. Yield of cowpea-maize, cowpea-okra, maize-cowpea-okra intercropping represent, respectively, 57.58%, 53.33%, 32.73% of cowpea seed yield recorded in monoculture.

Results found in 2016 at Gouna showed that, among the unsprayed, cowpea associated to maize recorded a higher seed yield followed by cowpea monocropped, cowpea in association to okra, and cowpea-maize-okra intercropping, but this difference was statistically not significant. In the sprayed, intercropping significantly reduced the cowpea seed yield. The yield of cowpea-maize, cowpea-okra, and maize-cowpea-okra intercropping represent, respectively, 52.66%, 59.76%, 34.91% of cowpea seed yield recorded in monoculture.

During the 2017 cropping years, among the unsprayed at Dang, the higher cowpea seed yield was observed on cowpea monocropped followed by cowpea-maize, cowpea-okra, and cowpea-maize-okra intercropping. From the sprayed in the same sites, yield of cowpea-maize, cowpea-okra, and maize-cowpea-okra intercropping represent, respectively, 52.66%, 59.76%, 34.91% of cowpea seed yield recorded in monoculture.

From the unsprayed in Gouna (2017), cowpea associated to maize resulted in a higher seed yield, followed by cowpea monocropped, cowpea-okra, and cowpea-maize-okra intercropping. The higher cowpea seed yield recorded on cowpea associated to maize could be the result of lower insect pest infestation in intercropping than in monoculture. In intercropping, shading, high humidity, and lower temperature effects were reported to keep cowpea insect pests' population low [35]. From the sprayed trials, yield of cowpea

seed yield recorded on cowpea-maize, cowpea-okra, and maize-cowpea-okra intercropping represent, respectively, 56.10%, 54.88%, and 36.59% of cowpea seed yield recorded in monoculture.

Like the result observed in sprayed, higher grain yield under monocropped cowpea compared to intercropping was reported by Chemeda [36]. Competition for water and shading are probably the two factors that reduced cowpea yield under high numbers of plants in intercrop [37]. This agrees with the report of Ofori and Stern [38] in maize and cowpea intercrops.

### 3.1.3. Okra

Table 5 presents the okra fruit yield as influenced by Cypermethrin treatment and intercropping system in 2016 and 2017 at Dang and Gouna. The comparison of okra fruit yield recorded in monoculture, sprayed to unsprayed, showed that insect pests reduced okra fruit yield by 24.72% and 43.86% in Dang and 23.25% and 27.51% in Gouna, respectively, in 2016 and 2017. Similar yield losses due to insect pests were reported to range between 15.47 to 56.45% in India [39].

**Table 5.** Okra fruit yield (t ha$^{-1}$) as influenced by Cypermethrin treatments and intercropping system in 2016 and 2017 at Dang and Gouna.

| Treatments | Dang 2016 | | | Dang 2017 | | |
|---|---|---|---|---|---|---|
| | Unsprayed | Sprayed | *t* | Unsprayed | Sprayed | *t* |
| O | 3.99 ± 0.28 [a] | 5.30 ± 0.34 [a] | 3.01 ** | 3.84 ± 0.32 [a] | 6.84 ± 0.35 [a] | 6.33 *** |
| MO | 3.50 ± 0.17 [ab] | 3.98 ± 0.23 [bc] | 2.97 ** | 2.96 ± 0.18 [b] | 4.65 ± 0.23 [b] | 5.76 *** |
| OC | 2.83 ± 0.34 [bc] | 4.83 ± 0.24 [ab] | 3.16 ** | 3.96 ± 0.27 [a] | 6.25 ± 0.26 [a] | 6.11 *** |
| MCO | 2.18 ± 0.18 [c] | 3.14 ± 0.29 [c] | 3.04 ** | 2.60 ± 0.20 [c] | 3.57 ± 0.24 [c] | 2.69 * |
| Mean | 3.12 ± 0.12 | 3.72 ± 0.11 | 5.80 *** | 3.34 ± 0.13 | 4.33 ± 0.12 | 9.25 *** |
| $F_{(3, 156)}$ | 12.47 *** | 9.97 *** | | 7.23 *** | 29.82 *** | |
| Treatments | Gouna 2016 | | | Gouna 2017 | | |
| | Unsprayed | Sprayed | *t* | Unsprayed | Sprayed | *t* |
| O | 3.83 ± 0.35 [a] | 4.99 ± 0.28 [a] | 2.57 ** | 3.90 ± 0.26 [a] | 5.38 ± 0.36 [a] | 3.13 ** |
| MO | 3.45 ± 0.26 [a] | 4.20 ± 0.20 [a] | 3.11 *** | 3.04 ± 0.26 [b] | 3.52 ± 0.20 [b] | 1.48 |
| OC | 2.96 ± 0.35 [b] | 4.93 ± 0.33 [a] | 3.05 ** | 4.03 ± 0.34 [a] | 5.27 ± 0.33 [a] | 2.60 * |
| MCO | 2.47 ± 0.22 [c] | 2.45 ± 0.21 [b] | 1.05 | 2.22 ± 0.22 [c] | 2.45 ± 0.22 [c] | 0.00 |
| Mean | 3.29 ± 0.15 | 3.36 ± 0.14 | 2.67 ** | 3.30 ± 0.15 | 3.69 ± 0.12 | 3.43 ** |
| $F_{(3, 156)}$ | 7.93 *** | 19.02 *** | | 9.00 *** | 21.72 *** | |

O: okra; MO: maize + okra; OC: okra + cowpea; MCO: maize + cowpea + okra. Results are expressed as mean (±) standard error. In column, means followed by same letter are not significantly different at 5%. *: $p < 0.05$; **: $p < 0.01$; ***: $p < 0.001$.

In 2016 and 2017 at Dang, overall, the higher okra fruit yield was observed on okra monocropped sprayed whereas the lower was on okra-maize-cowpea intercropping unsprayed. Among the unsprayed plot, okra-cowpea and okra-maize-cowpea intercropping significantly reduced the okra fruit yield compared to okra in monoculture in 2016. But in 2017, only okra-maize-cowpea intercropping reduced significantly the okra fruit yield compared to okra in monoculture. Among the sprayed in 2016, okra monocropped recorded a significant higher fruit yield compared to okra in intercropping. In 2017, there was not a significant difference between okra fruit yield recorded on okra in monoculture and okra associated to cowpea. Yield of okra-maize, okra-cowpea, and maize-cowpea-okra intercropping represent, respectively, 75.09%, 91.13%, and 59.25% in 2016 and 67.98%, 91.37%, and 52.19% in 2017 of okra yield recorded in monoculture.

In 2016 and 2017 at Gouna, the higher okra fruit yield was observed on okra monocropped sprayed whereas the lower was on okra-maize-cowpea intercropping unsprayed. Among the unsprayed plot, okra-maize-cowpea intercropping significantly reduced the okra fruit yield compared to okra in monoculture. The same trend was observed among the sprayed.

Among the sprayed, yield of okra-maize, okra-cowpea, maize-cowpea-okra intercropping represent, respectively, 84.17%, 98.80%, and 49.10% in 2016; 65.43%, 97.96%, and 45.54% in 2017 of okra yield recorded in monoculture. The results of this study corroborate the finding of Ajayi et al. [40] that okra-cowpea and okra-groundnut reduce okra yield compared to monoculture. Monocropped okra sprayed gave the highest fruit yield compared to intercropping sprayed because of less inter-specific competition among the crops as well as higher aggregate population density per unit area observed in the intercrop. This result is in disagreement with Hamma et al. [17] who reported that monocropped okra produced significantly lower yields compared to intercrops.

### 3.2. Land Equivalent Ratio

Table 6 presents the Land Equivalent Ratio. It appears from this table that all the intercropping favors the yield of the species (*LER* greater than 1). Except the cowpea-okra intercropping in 2016, the *LER* for the unsprayed plots is higher than those of their respective corresponding sprayed plot.

**Table 6.** Land equivalent ratio at Dang and Gouna.

| Treatments | Dang 2016 | Gouna 2016 | Dang 17 | Gouna 2017 |
|:---:|:---:|:---:|:---:|:---:|
| MC | 1.99 | 1.91 | 1.76 | 2.02 |
| MC + I | 1.40 | 1.41 | 1.41 | 1.49 |
| MO | 1.67 | 1.73 | 1.75 | 1.71 |
| MO + I | 1.61 | 1.62 | 1.56 | 1.43 |
| OC | 1.32 | 1.62 | 1.77 | 1.81 |
| OC + I | 1.79 | 1.51 | 1.51 | 1.53 |
| MCO | 1.81 | 2.00 | 2.19 | 1.84 |
| MCO + I | 1.48 | 1.46 | 1.60 | 1.59 |

MC: maize-cowpea; MO: maize + okra; OC: okra + cowpea; MCO: maize + cowpea + okra; + I: with insecticide.

In 2016, among the unsprayed, the higher *LER* was observed on maize-cowpea and maize-cowpea-okra intercropping at Dang and Gouna, respectively. In contrast, for okra-cowpea intercropping we found the lower *LER* on the both sites. Among the sprayed for the two sites, maize-okra intercropping had a higher *LER* whereas the lower was on maize-cowpea intercropping. In 2017, the higher *LER* was observed on maize-cowpea-okra and maize-cowpea intercropping at Dang and Gouna, respectively. The lower values were observed on maize-okra for the both sites. Among the sprayed, the higher *LER* were observed maize-cowpea-okra intercropping for the both sites. The lower was observed on maize-cowpea and maize-okra at Dang and Gouna, respectively.

Intercropping systems that consistently result in *LER* greater than one are thought to be more efficient systems, from a land use perspective, than monocrops [40]. Maize-cowpea intercrops have frequently out yielded monocropped maize or cowpea in many areas of the world. Legume/cereal intercropping has been shown as a practical method to conserve soil and to increase economic returns [41,42]. Similarly, Ijoyah and Dzer [18] reported that, intercropping okra with maize at the same time not only produced the lowest competitive pressure but gave the highest *LER* of 1.75.

### 3.3. Cost and Benefit

The cost and benefit of cropping systems with or without application of chemicals to avoid the development of insect pests or combat a pest when it has evolved requires detailed and case specific analyses. Although not an objective of our study, we consider on the cost side the acquisition of chemicals, labor, amount, quality of crops, and the negative effects on the environment and health. The benefits are good crops, which, however, are not ensured in the long run due to negative environmental effect. Several studies indicate a high cost of pesticide application, when externalities are considered [43,44]. However, these do not necessarily become visible to the producer. A targeted subsidizing system

could increase awareness and willingness of farmers and reduce cost, on the farm level but also on the societal level.

## 4. Conclusions

Our study was conducted in the Guinean Savannah and Sudano Sahelian agroecological zones of Cameroon. When this study began, very little was known about the effect of intercropping and Cypermethrin application on the yield in northern Cameroon. Our study revealed the benefits and limits of intercropping in both agro-ecological zones. Overall maize seed yield was higher at Gouna than at Dang during the both cropping seasons. The yield of cowpea and okra in 2016 were similar at Gouna and Dang. In 2017, lower okra fruit yield was observed at Gouna than at Dang. In monocultures, Cypermethrin significantly increased the yields indicating that unprotected plots suffered more from the activity of insect pests which results in the decreasing seed yield. Depending to the agroecological zones and years, reduction of yield due to insect pests ranged between 5–24%, 47–63%, and 23–44%, respectively, for maize, cowpea, and okra. In unsprayed, maize-cowpea intercropping increased slightly the maize seed yield compared to maize monocropped at Dang where higher infestation was observed. In 2016 and 2017 at Gouna, slightly higher yield of cowpea was recorded on cowpea-maize intercropping compared to cowpea in monoculture. Similar results were observed for maize yield on maize-cowpea intercropping in 2016 at Dang. Reduction of yield was observed in intercropping except for maize seed yield when growing in association with cowpea at Dang. Okra-maize-cowpea intercropping significantly reduced the yield due to their competitions on different resources. In addition, this type of intercropping, which contains more than two plants, is hard to weed. Considering the total yields obtained from each intercropping, intercropping trials resulted in generally higher yields compared to monoculture (*LER* > 1) in the both sites and years. The higher yield observed on monoculture sprayed than in monoculture unsprayed resulted in a slightly lower *LER* in sprayed. On the basis of the results obtained, we recommend maize-cowpea intercropping as a local sustainable solution to obtain more productivity.

**Author Contributions:** This work was carried out in collaboration between all authors. All authors have read and agreed to the published version of the manuscript.

**Funding:** We acknowledge the EU ERASMUS+ program for the financial support.

**Institutional Review Board Statement:** Not applicable.

**Informed Consent Statement:** Informed consent was obtained from all subjects involved in the study.

**Data Availability Statement:** Data are available on request from the first author.

**Acknowledgments:** We sincerely thank the reviewers for their important comments. We thank also Andreas Suchopar, technical assistant at the University of Bremen for his help for the soil analysis and Ndouyang Christian for the statistical analysis. Finally, we thank the team of the Ecology Department (J. Filser) of the Centre for Environmental Research and Sustainable Technology (UFT) of the University of Bremen for lab space and help.

**Conflicts of Interest:** The authors declare no conflict of interest.

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
