# Peer review of "Comparative Experimental Effects of Intercropping and Cypermethrin on Insect Pest Infestation and Yield of Maize, Cowpea and Okra in Two Cameroonian Agro-Ecological Zones"

_agriengineering, doi:10.3390/agriengineering3020025_

Round 1

Reviewer 1 Report

Please read the comments carefully and respond to the corrections and comments in the manuscript accordingly. Also please include line numbers in the manuscript.

Author Response

We thank you for your comments and remarks.

Corrections have been done throughout the manuscript according to your remarks and suggestions. There are some modifications in this new manuscript according to others reviewer's suggestions.  

References are arranged as per the journal specifications.

Reviewer 2 Report

For a start, I want to congratulate the authors for their work.

I have a number of suggestions:

  1. in figures 1 and 2 we have the notations NGA and GAR (refer to locations). please pass these notations (NGA and GAR) and in the first paragraph from point 2.1.  
  2. in figures 1 and 2 please put the notations on the vertical axes, precipitation, and temperature
  3. on 2.4 on the second proposition - cultivated species- you have pea nut or peanut??
  4. the conclusions can be extended

Author Response

We thank your for your comments. As you suggested, the following reference in the p. 2, line 6: GOMBAČ & TRDAN, 2014. The efficacy of intercropping with birdsfoot trefoil and summer savoury in reducing damage inflicted by onion thrips (Thrips tabaci, Thysanoptera, Thripidae) on four leek cultivars. Journal of plant diseases and protection, 121, 3: 117-124 has been added.

The conclusion has been extended.

Reviewer 3 Report

This is an excellent paper for which I suggest publishing, since the paper is written according to the rules for scientific paper. The authors used suitable methods, statistical analysis and important references from the field of study. Before acceptation for publishing I only recommend to add the following reference in the p. 2, line 6: GOMBAÄŒ & TRDAN, 2014. The efficacy of intercropping with birdsfoot trefoil and summer savoury in reducing damage inflicted by onion thrips (Thrips tabaci , Thysanoptera, Thripidae) on four leek cultivars. Journal of plant diseases and protection, 121, 3: 117-124. 

Author Response

Thank you for your comment and suggestions.

Corrections have been done throughout the manuscript. 

Figures 1 and 2 are deleted according to other reviewer’s suggestions. 

Data regarding the percentage of damaged plant are deleted after the new statistical analysis as recommended by other reviewer.

As you suggested, the following reference in the p. 2, line 6: GOMBAÄŒ & TRDAN, 2014. The efficacy of intercropping with birdsfoot trefoil and summer savoury in reducing damage inflicted by onion thrips (Thrips tabaci, Thysanoptera, Thripidae) on four leek cultivars. Journal of plant diseases and protection, 121, 3: 117-124 has been added. 

Reviewer 4 Report

The study seemed about leave pests only, which won't show the benefit of intercropping. Although the split-plot design was approximated implemented, the scale was far too small - just 50-75 plants per treatment. In addition, weekly application of Cypermethrin will definitely remove most insect pests. Thus leave damages caused by insects should be less and yields should be higher.

Intercropping usually serves two purposes: 1) attract insect pests and thus reduce the infestation to the main crop; 2) serve natural enemies as refugee so that there will be large number of them ready for the main crop. So from your experiment design, neither these impacts can be assessed (scale too small).

The the data analysis and presentation, can I suggest you consult with a statistician? For split-plot design, it requires more complicated data processing than Two-way ANOVA, and SPSS should be capable. By the way, since you have SPSS, why do you bother to use XLSTAT for two-sample t-test?

May I suggest you:

2. Materials and Methods

2.2 Precipitation: remove Fig.1 - as you didn't use month's rainfall in your data analysis. A summary indicates difference between to places will do.

2.3 Temperature: same as rainfall, Fig.2 is not necessary.

2.4 Soil and vegetation near the study sites: Table 1 is not necessary as well.

2.5 Experimental set-up: showing your design diagram for both randomised blocks and sub-plots. As I thought above, I am afraid these treatment sizes are far too small to show any effects.

By the way, you referred to "Fig. 31", where is it? Again, please judge if it is necessary for your current study.

2.8.1 Maize: don't you think this method is confusion? If like your design, each monoculture had 50 plants, you sampled 40 of them (that's reasonable in case any death), should you convert the result to t/ha according to this plant density? So for intercropping, otherwise the monoculture will definitely has higher yield.

2.8.2 Cowpea: any insects attracted to flowers, beans and they may not cause any leave damage.

2.8.3 Okra: again, considering insect pests to flowers and fruits.

3. Results and Discussion: please consult with a statistician and present results accordingly.

Please pay attention to the relationship of yield and leave damage. Sometime some leave damage may stimulate the crop and enhance product.

For the LER results, did you think about the reason why the treatment of insecticide has a few lower figures?

With proper data analysis, I hope you can tell the insecticide impacts, as well as cropping system effects, which should be comparable between both places.

Author Response

We thank you a lot for your comments and suggestions.

The present study aims to compare the efficiency of intercropping to that of Cypermethrin application and to evaluate their effect on the yield of maize, cowpea and okra. Some corrections have been done throughout the manuscript. In the materials and methods, Figures showing the rainfall and temperature have been removed as suggested. Data were collected insect damage to stem, flowers, cob/pod/fruits of each of the three crops. Paper has been published on the diversity of potential insect pests associated to the three plants tested in 2019.  In the previous manuscript submitted, we have presented the results on the number of damaged plants and crop yield, being regarded as the most integrating and indicative parameters. Following your remarks and after the new statistical analysis we judged necessary to present only the results of crop yields. Cypermethrin was used as positive control as reported in many studies. We agree that more frequent application would remove most insects. However, the frequency chosen is in agreement with the practice in the Northern parts of Cameroon. The comparison of yield recorded on sprayed plots to that of unsprayed conducted us to have an idea about the approximate yield loss due to insect activities on maize, cowpea and okra in the both years and sites. The comparison of yield recorded on monoculture sprayed to that of intercropping sprayed lead us to have an idea on the approximate yield loss due to the effect of completion between plants when growing in intercropping.

In accordance to your remark, intercropping usually serves two purposes: 1) attract insect pests and thus reduce the infestation to the main crop; 2) serve natural enemies as refugee so that there will be large number of them ready for the main crop. Although from our experiment design the scale is small, for the reasons mentioned before we consider it valid to establish at least in part the interactions mentioned in the comment. Also some previous study using similar scale have been published. Results obtained by these authors shown the interaction between plant and insect pests when plants are growing in the same scale.

Yield of the plant was obtained on 40 randomly selected plants per treatment. Concerning maize for instance, each treatment has a total of 200 plants per treatment not 50 as reported in the comment (there was 4 repetitions for each treatments). For okra and cowpea, we have more than 200 plants per treatment. This sampling is by far representative to obtain a good results.

Concerning the data analysis and presentation, new statistical analysis have been done by the help of statistician. Data were analyzed using SPSS software. Tukey test was used for the separation of more than two means. Student t test was used to compared the yield recorded on unsprayed to that of sprayed.

The results are presented according to the method of Kwanchai and Gomez, 1984 in statistical procedures for agricultural research 2nd edition, 704p.

In the results presented the insecticide impacts as well as cropping system is clear.  For the LER results, the treatment of insecticide has a few lower LER simply because yield on monoculture sprayed was by far higher than yield obtained in monoculture unsprayed.

Round 2

Reviewer 1 Report

Dear Authors, 

I believe you did not perform all the changes that were suggested in my previous comments. Please check the new version with comments and rectify them as well as please correct the previous ones. Please elaborate on the discussion part. You need to demonstrate that your findings are in line or contradicting other research work.

Good luck!

Author Response

We thank you again for your remarks and suggestions.

In order to improve the manuscript, corrections have been done throughout.

Some addings and modifications have been done throughout the document, particularly in the result and discussion sections according to your suggestion in the previous manuscript.

References are arranged as indicated in author guidelines.

Reviewer 4 Report

Thanks for your efforts. I've put some marks, comments/suggestions in the attached pdf.

Author Response

We thank you again for your remarks and suggestions.

In order to improve the manuscript, corrections have been done throughout.

Some addings and modifications have been done throughout the document, particularly in the result and discussion sections.

References are arranged as indicated in author guidelines.